# Exploring the impact of primer length on efficient gene detection via high-throughput sequencing

Julia Micheel [1,2], Aram Safrastyan [1,2], Franziska Aron [1,2] & Damian Wollny [1,2,3] ✉

Reverse transcription (RT) is a crucial step in most RNA analysis methods. Optimizing protocols for this initial stage is critical for effective target detection, particularly when working with limited input RNA. Several factors, such as the input material quality and reaction conditions, influence RT efficiency. However, the effect of RT primer length on gene detection efficiency remains largely unknown. Thus, we investigate its impact by generating RNA-seq libraries with random RT primers of 6, 12, 18, or 24 nucleotides. To our surprise, the 18mer primer shows superior efficiency in overall transcript detection compared to the commonly used 6mer primer, especially in detecting longer RNA transcripts in complex human tissue samples. This study highlights the critical role of primer length in RT efficiency, which has significant potential to benefit various transcriptomic assays, from basic research to clinical diagnostics, given the central role of RT in RNA-related analyses.

RNA analysis is a crucial aspect of gene expression studies with great relevance from fundamental research to clinical diagnostics. The initial step for the vast majority of RNA analyses is to transcribe the RNA into cDNA. This reverse transcription (RT) step is of great significance, because inefficient RT has been shown to be a major bottleneck for transcriptomic analysis[1–5]. Consequently, significant efforts have been made in recent years to enhance the efficiency of RT[1–3,5–11]. Many factors are crucial for the efficiency of RT such as quality of the input material, priming strategy, the RT enzyme, buffer, and reaction conditions. In all these areas, a lot of optimization work has already been done to improve RT efficiency, e.g., by the introduction of molecular crowding reagents and optimization of incubation temperatures[12–14]. Optimizing the reverse transcription process is especially critical in applications where a comprehensive representation of the transcriptome is desired, such as total RNA sequencing, or when working with limited input RNA is desired, e.g., in single-cell analyses or liquid biopsies.

In terms of priming, oligodT has become widely established for the selective detection of polyadenylated RNAs although it presents difficulties with fragmented samples or RNAs that contain strong secondary structures[3]. This is because the polyA tail is only found at

the 3' end and thus, in case of fragmentation, only the 3' fragment has a binding site for reverse transcription. In the case of strong secondary structures, they can hinder reverse transcription and thereby cause reverse transcription termination. In this case, again only the 3' end will be reverse transcribed[3,15,16].

In contrast to exclusive analysis of polyadenylated transcripts, when the objective is to analyze the entire transcriptome, there are two main options: in vitro polyadenylation of all RNAs present and subsequent oligodT priming, as well as the widely used random priming. In addition, random priming can increase transcript coverage of fragmented samples or when continuous RT is hindered by extensive secondary structures of the template[17]. The most commonly used primer for random priming is the random 6mer. Sporadically, studies chose random primers of different lengths (Fig. 1A)[15,18–23]. However, the impact of random primer length on the quality of RNA sequencing results remains unknown. Here, we compare the performance of random primers with varying length in the generation of RNA-seq libraries from human brain total RNA. Surprisingly, our results suggest that the random 6mer does not yield the best overall performance. Instead, the random 18mer primer is found to be more efficient in detecting most

[1]RNA Bioinformatics and High Throughput Analysis, Friedrich Schiller University, Jena, Germany. [2]Leibniz Institute on Aging—Fritz Lipmann Institute (FLI), Jena, Germany. [3]Max Planck Institute for Evolutionary Anthropology, Leipzig, Germany. ✉e-mail: labwollny@gmail.com

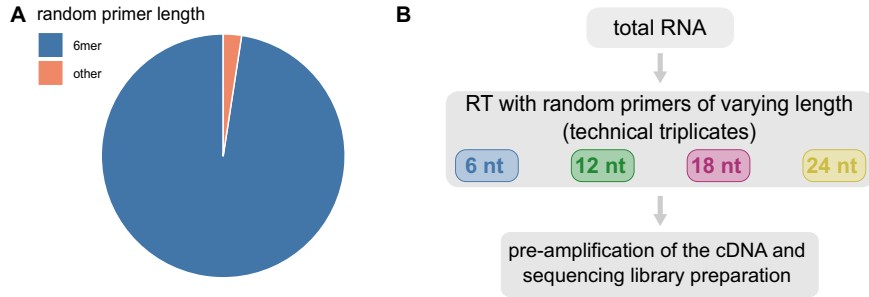

**Fig. 1 | Rationale and experimental design of the study. A** The pie chart visualizes the number of publications using random 6mers compared to the publications describing the use of random primers of other lengths between 4 and 24 nucleotides. **B** Schematic of the experimental workflow from RNA input to cDNA synthesis with random primers of different length in technical triplicates, cDNA pre-amplification, and sequencing library generation. Source data is provided as a Source Data file.

genes. In particular, longer transcripts such as protein-coding and long non-coding RNAs are detected more efficiently from human tissue samples, while detection of shorter RNAs is not as strongly affected.

## Results

### Primer length effects on library complexity

In order to test how random primers of different lengths would affect RNA sequencing results, we prepared RNA sequencing libraries based on the SMART-seq3 library preparation protocol in technical triplicates[14]. We implemented random primers of 6, 12, 18, and 24 nucleotides lengths at the reverse transcription step of the protocol (Fig. 1B, Supplementary Fig. 1). TapeStation results of the subsequently obtained cDNAs prior to fragmentation indicated no visual differences in the size distribution of the libraries as a result of different RT primer lengths (Supplementary Fig. 1).

After Illumina sequencing of the libraries, we aimed to compare the complexity of the libraries generated with random primers of different lengths. To ensure unbiased comparison, we conducted random subsampling (5 million reads) of the raw reads to obtain equal read sets. We found that the random 18mer primer detected both the most genes (Fig. 2A, Supplementary Data 1) and the most transcripts (Fig. 2B, Supplementary Data 1). The efficiency of gene detection by the 18mer compared to the other primers was particularly pronounced for lowly expressed genes (FPKM 1–20, Fig. 2C, Supplementary Data 1). Furthermore, this effect occurred regardless of the degree to which we subsampled, but it was more pronounced at high sequencing depth (Fig. 2D, Supplementary Data 1). Notably, the 18mer primer identified on average the same amount of genes with only 2.5 million analyzed reads compared to 5 or 10 million reads generated with 6mer, 12mer, or 24mer primers (Fig. 2D). We also analyzed the reliability of gene detection across technical replicates and found high reliability irrespective of the primer chosen (Fig. 2E).

Given the notable differences in the number of genes detected, we investigated how large the intersections of the detected genes are and how many genes can only be detected uniquely by one of the primers. While the majority (5682 of 13,298 genes) was detected across all primers, primer length specific genes were detected as well. The fraction of genes which were only detected for a given primer was between 4 and 5% for the 6mer, 24mer, and 12mer, while 10% of genes were detected by the 18mer only (Fig. 2F, Supplementary Data 3–6).

### Characterization of the detected genes

Next, we analyzed the characteristics of genes that were detected with the different random primers. Overall, the same gene biotypes were found in each library. They mainly consist of protein-coding genes regarding the number of detected genes (Fig. 3A) as well as the read counts (Fig. 3B). The 18mer samples detected the highest number of

protein-coding genes, whereas the 12mer, 24mer, and 6mer primers detected similar amounts, with the 6mer primer being the least efficient (Fig. 3A). Similarly, other RNA biotypes comprising long genes such as lncRNAs or pseudogenes were detected most efficiently by the 18mer primer (Fig. 3A, Supplementary Data 1). On the other hand, there was a tendency for short RNA biotypes (snRNAs and snoRNAs) to be better detected with shorter primers, however future experiments using dedicated short RNA sequencing libraries will be needed to explore this effect further (Supplementary Data 1). This length-dependent distribution of detected genes was even more apparent when genes were classified based on transcript lengths rather than biotypes (Fig. 3C, Supplementary Data 1). This indicates that more efficient detection of long genes might be the main cause for the higher efficiency of gene detection by the 18mer in our study. We also classified the transcripts based on their GC content and examined whether certain primers demonstrate better performance in detecting transcripts of a specific GC content. While no primer showed a notable advantage in detecting transcripts with lower GC content (20–40%), transcripts with higher GC content tended to be detected more efficiently using the random 18mer which was significantly pronounced within the GC range of 60 to 80% (Fig. 3D, Supplementary Data 1).

Given the fact that the 18mer detects substantially more genes than the commonly used 6mer, one potential concern is that some of these genes may represent artifacts, e.g., due to higher degree of read misalignment in the 18mer datasets[24]. To address this, we conducted an analysis to ascertain whether the detected genes are indeed expressed in the brain, since our input material was RNA isolated from human brain samples. The Human Protein Atlas reports that a total of 16,465 protein-coding genes have been detected in the brain, which encompasses both brain-specific and housekeeping genes[25]. Notably, more than 97% of the genes detected using the random 18mer match this list (Fig. 3E). Likewise, the protein-coding genes detected with the other random primers were 97%–98% consistent with those expressed in the brain (Supplementary Table 1).

When we only take the uniquely detected genes into account, the 18mer enables detection of 979 more protein-coding genes (95%), while only 47 protein-coding genes were not reported in the Human Protein Atlas (Fig. 3F, Supplementary Table 2). For the other primers, 94 to 98% of the 315 to 413 uniquely detected protein-coding genes were in alignment with the Human Protein Atlas (Fig. 3F, Supplementary Table 2). In order to further confirm this result, we performed tissue enrichment analysis of the identified protein-coding genes using tissue-specific genes defined by the Human Protein Atlas[26]. Significant enrichment of cerebral cortex specific genes was detected but no significant detection of genes enriched in any other tissue could be observed (Supplementary Fig. 2). In addition, we performed gene enrichment analysis of the uniquely detected genes per primer length

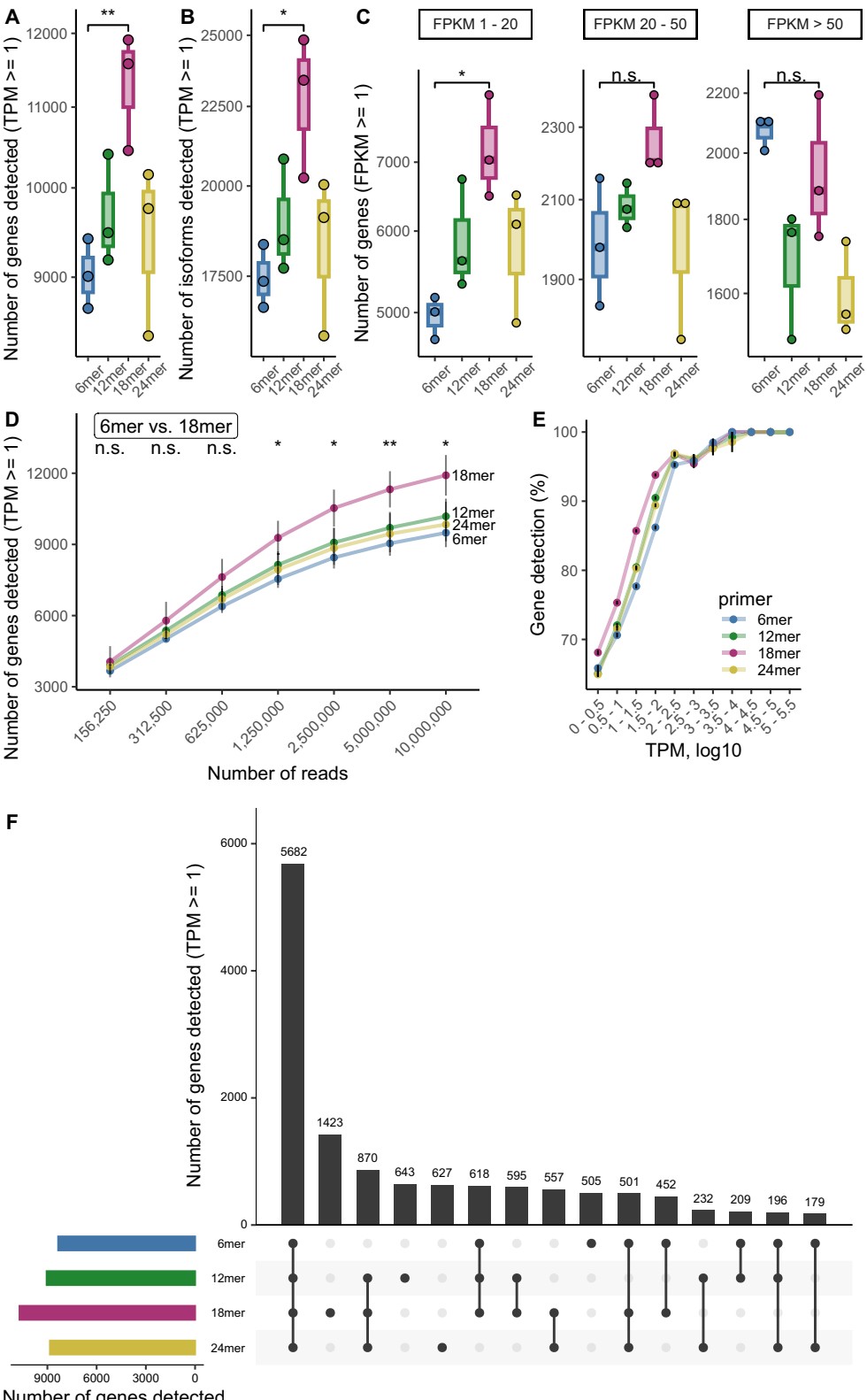

(Supplementary Data 3–6) to see if there was a biological pattern. Only a fraction of the genes (0 to a maximum of 55%) can be assigned to significantly enriched pathways (Supplementary Table 3). The significantly enriched pathways are highly variable (Supplementary Data 7) and no general biological pattern is observed. This result confirms that the detection of unique genes is a technical effect and not a biological one.

### Verification of the result reproducibility

To assess the reproducibility of our results, we conducted a second independent experiment at substantially higher sequencing depth using the same experimental setup as in the first experiment (Fig. 1B). All replicates of this experiment were computationally subsampled to 30 million reads. In accordance with our first experiment, we noticed an enhanced detection of total genes (Supplementary Fig. 3a) and to a

**Fig. 2 | Quantification of detected genes depending on the random primer length. A** Total human brain RNA sequencing libraries were generated in technical triplicates using random primers of 6, 12, 18 or 24 nucleotides. After sequencing, the libraries were computationally subsampled to 5 million reads per sample for comparison of the library complexity dependent on random primer length. The numbers of genes detected (TPM > = 1) and (**B**) the numbers of isoforms detected (TPM > = 1) from 5 million raw reads per sample are depicted ($n$ = 3 technical replicates). **C** The numbers of detected genes (FPKM > = 1) per FPKM bin are depicted as box plots (5 million reads, $n$ = 3 technical replicates). **D** Quantification of gene detection after random subsampling (156,250–10 million raw reads) is shown ($n$ = 3 technical replicates). Error bars = mean ± standard deviation. **E** Percentage of gene detection reproducibility across replicates (5 million reads,

$n$ = 3 technical replicates) in dependence of gene expression levels (TPM). Error bars = mean ± standard error of the mean. **F** Intersections of the genes detected using the random primers of different lengths after computational subsampling to 5 million reads (TPM > = 1). Here, all genes were considered per primer that were found in at least two of three replicates. The total numbers of genes detected per random primer fulfilling this criteria are depicted as an additional bar chart on the left. Box plots display the interquartile range (25th–75th percentiles); center line = median; whiskers extend to the largest (maxima) and smallest (minima) values within 1.5 x interquartile range. *$p$ < 0.05, **$p$ < 0.01, n.s. = not significant. Source data is provided as a Source Data file. Statistical tests (unpaired two-sided Student's $t$ test and Mann–Whitney $U$-test) performed in (**A–D**) with corresponding $p$ values are reported in Supplementary Data 1.

lesser extent of transcripts (Supplementary Fig. 3b) by the random 18mer (Supplementary Data 1). Furthermore, the number of genes detected per number of reads was best for 18mer, independent of subsample size (Supplementary Fig. 3c, Supplementary Data 1) and with the 18mer we again detected a big proportion of unique genes (Supplementary Fig. 3d, Supplementary Data 8–11). The efficiency of gene detection for lowly expressed genes was higher for the 18mer compared to the other primers (with exception of one replicate of the 12mer) (FPKM 1–20, Supplementary Fig. 3e, Supplementary Data 1). The distribution of detected genes per biotype (Supplementary Fig. 4a) and TPM per biotype (Supplementary Fig. 4b) was similar to that of the first experiment (Fig. 3A, B), with protein-coding genes making up the largest proportion. Although the relative differences between the 6mer, 12mer, and 24mer primers shifted in comparison to the initial experiment conducted at less sequencing depth (Supplementary Figs. 3, 4), the second experiment confirmed the advantage of using the 18mer primer for detecting longer transcripts, whereas for shorter biotypes, the 6mer primer showed again an advantage (Supplementary Fig. 4a, c, Supplementary Data 1). Compared to the first sequencing run (Fig. 3B), there are no differences in the variability of the detected biotypes between 18mer and the other samples (Supplementary Fig. 4b). In addition, no relationship between GC content and detection efficiency of the different primer lengths could be detected (Supplementary Fig. 4d). These differences between the experiments could be explained by the different sequencing depth. Differences that were observed during superficial sequencing could be compensated by deeper sequencing.

**Primer-dependent RT efficiency across diverse input materials**
To further verify the primer-length-dependent effect on reverse transcription efficiency, we performed an additional sequencing experiment with a different RNA source as input. We prepared RNA sequencing libraries as described before from RNA which was isolated from Vero cells, an African Green Monkey cell line. All samples were computationally subsampled to 7 million reads. In line with the two independent experiments with human brain input RNA, the random 6mer didn't show the highest gene detection efficiency (Supplementary Fig. 5a). Again, we see that there is a tendency towards enhanced gene detection with longer random primers but the overall differences between the primers are not as big as in the human brain experiments and do not reach statistical significance (Supplementary Data 1). Consistent with the other experiments, there is a large intersection of genes that were detected with all random primers of different lengths, but also genes that were only uniquely detected with individual primers (Supplementary Fig. 5b). Most primer-specific genes were detected with the 18mer and the 24mer (Supplementary Fig. 5b). The libraries consist largely of protein-coding genes, both in terms of the number of genes detected and the proportions of reads (Supplementary Fig. 6a, b). The correlation between primer length and transcript length shown in the previous experiments can also be observed here, but to a lower extent: in the range up to 200 nt in length, we detect approximately the same number of transcripts with all primers (Supplementary Fig. 6c).

In order to further investigate the validity of our results across more variables we quantified the differences of reverse transcription efficiency under more defined conditions. To achieve that, we used artificial RNA of defined lengths of 50, 80, 150, 300, 500, and 1000 nt as input material. We then carried out reverse transcription comparing the 6mer and 18mer and quantified the generated cDNA on the fragment analyzer (Supplementary Fig. 7). In accordance with the results we obtained by RNA sequencing, we again found that shorter cDNAs tend to be more efficiently generated by the random 6mer while longer cDNAs are more efficiently generated by the random 18mer (Supplementary Fig. 7, Supplementary Table 4). We further tested if different RNA input amounts would affect this result and found that the effect was observed with 5 ng input RNA ($p$ = 0.00737) and to a smaller extent with 20 ng input RNA (not significant) (Supplementary Fig. 7). Lastly, we performed the same experiment with a mixture of both, the 6mer and the 18mer primers. Here, the beneficial effects of both primers can be observed in the fragment lengths of the generated cDNAs (Supplementary Fig. 7, Supplementary Table 4), indicating that a mixture of short and long primers enhances reverse transcription of RNAs with varying lengths.

Taken together, these results demonstrate the superior performance of the 18mer RT primer over the commonly used 6mer for the efficient detection of long transcripts (in particular > 1000 bp) by high-throughput RNA sequencing.

## Discussion
The importance of the RT as the first step of almost all assays aiming to analyze RNA motivated the community to continually advance the efficiency of this process, in particular for low input assays. Despite significant advancements in this area, the efficiency of cDNA synthesis is limited. Therefore, it is imperative to enhance this process in order to increase sensitivity especially when working with small input amounts of RNA. Our study focuses on the optimization of the priming strategy as a crucial, yet overlooked, factor in reverse transcription. Specifically, we examined the impact of random primers with varying lengths on the efficiency of reverse transcription, particularly concerning the complexity of the resulting library. To our surprise, we discovered that the random 18mer primers were the most effective in detecting different transcripts, particularly longer biotypes (>1000 nt) that constitute a significant proportion of the libraries. The reason for the efficiency advantages of the 18mer particularly for converting long RNAs into cDNA might be due to the increased number of hydrogen bonds between the primer and its target, which might not be of benefit for even longer primers due to potential secondary structure formation. Yet, more work is necessary to fully uncover the molecular mechanism underlying this effect. Irrespective of the underlying mechanism, the effect has proven to be robust across several conditions that we tested in the course of this study. Thus, from a practical point of view, replacing the random 6mer with the 18mer, or adding the 18mer to the 6mer, are easy implementations to enhance library complexity for transcriptomic studies. In addition, it is a very cost effective way as the additional cost is restricted to the cost of the

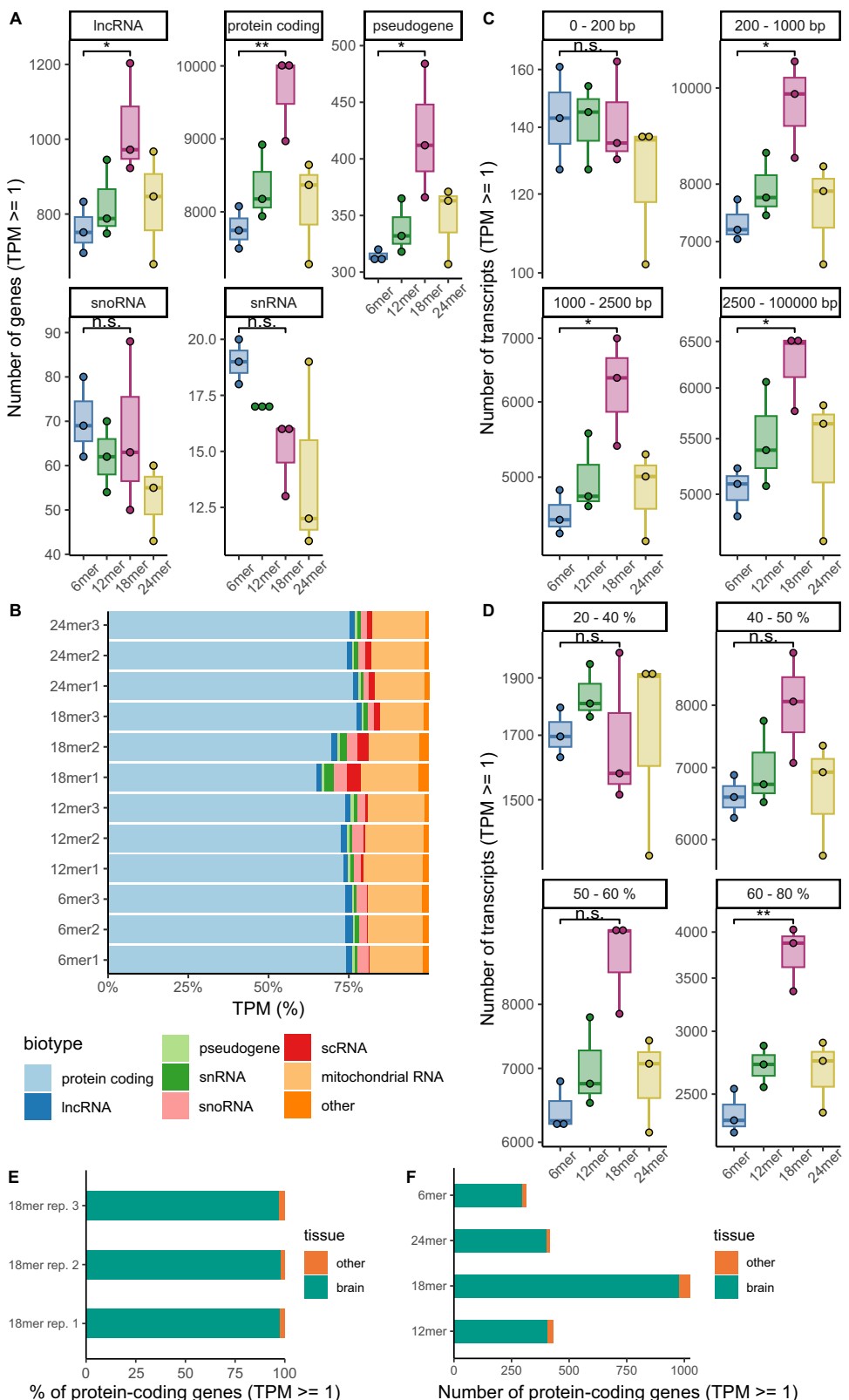

longer random primer which is about 33% more compared to the random 6mer and is overall negligible compared to the total cost of RNA-seq. Otherwise no additional steps or reagents are needed for implementation.

The utilization of random 6mers is widely established (Fig. 1A) as it is long enough to allow efficient template binding and a certain degree of randomness ($4^6$ possible nucleotide sequences) and it is fairly easy to synthesize. In addition, it is short in order to avoid formation of secondary structures within the primers. In contrast, there is limited literature on the use of longer random reverse transcription primers[15,18–23]. For instance, a previous microarray analysis predominantly identified various transcripts using a random 15mer primer[19]. While one publication supports this observation[23] by qPCR of single genes, another qPCR-based study contradicts it[22]. However,

**Fig. 3 | Characterization of detected genes by random primers of varying length. A** The numbers of detected genes (TPM > = 1) per biotype are depicted (5 million reads, $n$ = 3 technical replicates). **B** Transcript proportions of the different biotypes detected per sample (subsampled to 5 million reads). **C** The numbers of detected transcripts (TPM > = 1) per transcript length are depicted (5 million reads, $n$ = 3 technical replicates). **D** The numbers of detected transcripts (TPM > = 1) per transcript GC content are depicted as box plots (5 million reads, $n$ = 3 technical replicates). **E** Proportions of protein-coding genes that are expressed in brain or other tissues according to the Human Protein Atlas for the three technical replicates of the 18mer sample, subsampled to 5 million reads. **F** Numbers of primer-specific detected protein-coding genes that are expressed in brain or other tissues according to the Human Protein Atlas. Here, all genes were considered per primer that were found in at least two of the three replicates. Box plots display the interquartile range (25th–75th percentiles); center line = median; whiskers extend to the largest (maxima) and smallest (minima) values within 1.5 x interquartile range. *$p < 0.05$, **$p < 0.01$, n.s. = not significant. Source data is provided as a Source Data file. Statistical tests (unpaired two-sided Student's $t$ test and Mann–Whitney $U$-test) performed in (**A**–**D**) with corresponding $p$ values are reported in Supplementary Data 1.

neither of the latter two qPCR studies investigated the effect of RT primer length on a global scale. Instead, they focused exclusively on the detection of specific individual genes. As we have shown in the present study, the RT efficiency with the different primers depends on the transcript length. Therefore, to detect and quantify the effect of primer length on RT, an unbiased, highly sensitive analysis on a global scale using high throughput sequencing, as in our study, is required. The observed positive effect of the 18mer primer on the detection of longer transcripts might be particularly pronounced at low input levels (Supplementary Fig. 7), reflecting the range for which the protocol used is tailored. Increasing the input level in the fragment size experiment to 20 ng shows a lower, statistically insignificant effect compared to the random 6mer, which may indicate that the priming efficiency of the different primer lengths converges at higher input levels.

In addition, the Vero cell experiment showed a lower positive effect of the 18mer primer compared to the human brain experiments which did not reach statistical significance (Supplementary Data 1). One possible reason for this could be the lower complexity of the Vero cell samples. It is also notable that in the Vero cell samples there is considerable variance between the technical replicates per sample in the number of genes detected for the 6mer, 12mer, and 24mer primers. When we compared the total number of genes detected with the concentrations of cDNAs generated during reverse transcription and cDNA preamplification, we found a strong correlation (Supplementary Table 5). Therefore, these cDNA concentrations pre-sequencing could serve as a useful proxy to estimate the success of the reverse transcription reaction, which in our case was also consistently high for the 18mer primer (Supplementary Table 5).

Furthermore, it is important to consider that the observed effect may be protocol-specific. In our study, we employed the modern and highly optimized SMART-seq3 protocol, wherein the reverse transcription reaction occurred at 42 °C, with alternating cycles of 50 °C and 42 °C. These temperature variations were designed to diminish secondary structures in the template RNA. Concurrently, they may also contribute to reducing secondary structures of the random primers. Moreover, 18mer primers could potentially benefit from this optimization, as they can still bind more effectively to the template RNA at higher temperatures compared to shorter primers. In this context, it is important to note that the random primers have 25 nucleotide long 5' adapters (Table 1), which are not used for binding to the RNA target, but as a platform for priming in the subsequent preamplification step (Supplementary Fig. 1b). These adapters could have additional effects on the reaction kinetics. Although they are not always present in other RNA-seq library preparation kits, most of the modern, highly frequently used library preparation methods such as the Chromium Single Cell 3' chemistry, uses adapters for barcoding and priming which suggests that implementing the 18GTP (Thermo Fisher Scientific) might also be beneficial for these commonly used techniques.

In conclusion, our study highlights the significance of optimizing the priming strategy in reverse transcription to improve the efficiency of cDNA synthesis. Specifically, we found that the use of random 18mer primers was most effective in detecting a wide range of transcripts, particularly longer ones (>1000 nt). These findings suggest that primer length plays a role in reverse transcription efficiency and underscore the importance of unbiased approaches such as RNA-seq for comprehensive evaluation of the priming strategy.

## Methods
Detailed information on all consumables mentioned in the Methods section is provided in Supplementary Data 12.

### Sequencing library preparation
The sequencing library preparation was adapted from the Smart-seq3 protocol[14] but instead of the oligodT primer we implemented random primers with 6, 12, 18, and 24 nucleotides length for reverse transcription (Fig. 1B). Since the effect of primer length choice on gene detection is an understudied area, we aimed to investigate a wide range of primer lengths. We therefore opted for primer lengths that are a multiple of 6mer. We set the maximum primer length at 24 nucleotides, as increasing primer length increases the likelihood of secondary structure formation that would hinder the priming of transcripts or promote the formation of primer dimers[27]. As input material, 1 ng Human Brain Total RNA (Invitrogen, USA) or 1 ng RNA isolated from Vero cells (Cytion, Germany) infected with SARS-CoV-2 delta variant were used. The cell cultivation, infection, and RNA isolation are described in detail elsewhere[28]. In technical triplicates, the respective amount of input material diluted in 1 μl nuclease-free water was added to 3 μl lysis buffer containing 5% PEG BioUltra, 8000 (Sigma Aldrich, Germany), 0.1% Triton X-100 (Sigma Aldrich), 0.5 U μl$^{-1}$ Recombinant RNase Inhibitor (Takara, Japan), 0.5 mM dNTPs (Thermo Fisher Scientific, Germany) and 0.5 μM of the respective random primer (IDT, USA, sequences shown in Table 1). Incubation was performed at 72 °C for 10 min. Afterwards, 1 μl of reverse transcription mix was added, including 25 mM Tris-HCl (pH 8.0) (Sigma Aldrich), 30 mM sodium chloride (Invitrogen), 2.5 mM magnesium chloride (Invitrogen), 1 mM, 8 mM DTT (Invitrogen), 0.5 U μl$^{-1}$ Recombinant RNase Inhibitor, 2 U μl$^{-1}$ Maxima H Minus Reverse Transcriptase (Thermo Fisher Scientific) and 2 μM TSO (IDT, sequence shown in Table 1). For reverse transcription and template switching, the mixture was incubated at 42 °C for 90 min followed by 10 cycles of incubation at 50 °C and 42 °C for 2 min each and enzyme inactivation at 85 °C for 5 min. 6 μl pre-amplification mix containing 1x KAPA HiFi HotStart ReadyMix (Roche, Switzerland) and 0.1 μM ISPCR primer (IDT, sequence shown in Table 1) were added to the reaction afterwards followed by 98 °C 3 min initial denaturation, 20 cycles of 20 s denaturation at 98 °C, 30 s annealing at 65 °C and elongation at 72 °C for 4 min and a final elongation step at 72 °C for 5 min. The PCR product was cleaned up using SPRIselect beads (Beckman Coulter, USA) in a 1.2x ratio. After two washing steps with 80% EtOH p.a. (Thermo Fisher Scientific), the cDNA was eluted in 10 μl nuclease-free water. Samples were quantified by Qubit dsDNA High Sensitivity (Invitrogen) and library size distributions were analyzed with TapeStation using D5000 ScreenTape (Agilent, USA) or Bioanalyzer using the High Sensitivity DNA kit (Agilent). Exemplary TapeStation plots of the pre-amplified cDNAs generated with random primers of different lengths are shown in Supplementary Fig. 1a.

Tagmentation was performed with the Nextera XT Library Prep Kit (Illumina, USA). For this, cDNA was diluted to 0.3 ng μl$^{-1}$ and 1.25 μl

**Table 1 | DNA oligonucleotide details**

| Primer | Sequence (5' – 3') | References |
|---|---|---|
| random 6mer | AAGCAGTGGTATCAACGCAGAGTACN6 | adapted from[16] |
| random 12mer | AAGCAGTGGTATCAACGCAGAGTACN12 | adapted from[16] |
| random 18mer | AAGCAGTGGTATCAACGCAGAGTACN18 | adapted from[16] |
| random 24mer | AAGCAGTGGTATCAACGCAGAGTACN24 | adapted from[16] |
| TSO | Biosg/AAGCAGTGGTATCAACGCAGAGTACATrGrG+G | 16 |
| ISPCR primer | AAGCAGTGGTATCAACGCAGAGT | 16 |

Oligonucleotides used for RNA sequencing library generation with their respective sequences and original publications. Biosg/indicates 5' biotinylation, rG is the RNA-base guanine, +G is a locked guanine base, NX indicates random nucleotides and the respective number.

of the dilution were mixed with 2.5 μl Tagment DNA buffer and 1.25 μl Amplicon Tagmentation mix (Tn5). The mixture was incubated at 55 °C for 10 min. Afterwards, the Tn5 reaction was stopped by adding 1.25 μl NT buffer. 1.25 μl of each Nextera index primer (Illumina) were added as well as 3.75 μl Nextera PCR Master mix. Library amplification of the tagmented samples was carried out by incubation at 72 °C at 10 min and 30 s at 95 °C, followed by 12 cycles of 10 s denaturation at 95 °C, 30 s annealing at 55 °C and elongation for 60 s at 72 °C, as well as a final elongation step for 5 min at 72 °C. For clean up, SPRIselect beads were used in a 1x ratio. After two washing steps using 80% EtOH p.a., cDNA was eluted in 5 μl nuclease-free water and its concentration was quantified using the Qubit dsDNA HS. Library size distributions were analyzed using the TapeStation D5000 ScreenTape or the Bioanalyzer 2100 expert with the High Sensitivity DNA assay. After equimolar pooling, the libraries were sequenced on an NovaSeq6000 instrument (Illumina). The first human brain RNA experiment as well as the SARS-CoV-2-infected Vero cell RNA experiment were sequenced with a NovaSeq6000 SP 500 cycles v1.5 kit, paired-end (2 × 250 bp) and the second human brain RNA experiment with a NovaSeq6000 SP 300 cycles v1.5 kit, paired-end (2 × 150 bp).

### Sequencing data processing
In order to account for differences in sequencing depth, samples were randomly subsampled using the tool seqtk (version 1.0-r82-dirty)[29]. In the first experiment, samples were subsampled to a total of 10,000,000, 5,000,000, 2,500,000, 1,250,000, 625,000, 312,500, and 156,250 reads. In the second experiment, owing to an overall higher sequencing depth, samples were subsampled to a total of 30,000,000, 20,000,000, 15,000,000, 10,000,000, 5,000,000, and 2,500,000 reads.

Afterward, adapters were trimmed, and ribosomal reads were filtered by the BBDuk program of the BBMap suite of tools (version 38.93)[30]. Subsequently, samples were mapped to the human genome (build GRCh38) using the default parameters of the aligner STAR (version 2.7.8a) and the flag "--quantMode TranscriptomeSAM GeneCounts"[31]. Gene and transcript expression measured in transcripts per million (TPM) and fragments per kilobase of transcript per million fragments mapped (FPKM) values were calculated using the function "rsem-calculate-expression" of the RSEM tool (version 1.3.1)[32] with the parameters "--paired-end" and "--strandedness none" using the human genome annotation of the GENCODE consortium (build GRCh38 release 38). A schematic of the workflow from sequencing library generation to data analysis is depicted in Supplementary Fig. 1b.

For the analysis of the sequencing results of the African Green Monkey (AGM) Vero cells infected with the SARS-CoV-2 delta variant, samples were first subsampled to 7,000,000 total reads as already described. Afterwards, the adapters were trimmed from the raw reads as already described and subsequently mapped first to the AGM genome (assembly ChlSab 1.1) and then separately to the SARS-CoV-2 genome (assembly ASM985889v3) as already described. TPM values were calculated as already described. The AGM gene biotype information was extracted from the corresponding gene annotation file (assembly ChlSab 1.1).

R (version 4.2.2) was used for analysis and R package ggplot2 (version 3.4.2)[33] for the visualization of sequencing data. Gene names and biotype information, transcript length, and transcript GC content were retrieved via the R package ensembldb (version 2.20.2)[34] from Ensembl version 105 (annotation hub 'AH98047'). Transcripts were considered to be expressed with a TPM value of at least 1 or an FPKM value of at least 1. In most analyses, we show the individual data points of the technical replicates. To identify which genes were specifically detected with a particular primer length, we considered all genes that were detected in at least 2 of the 3 technical replicates. UpSet plots were generated with the R package UpSetR (version 1.4.0)[35]. Tissue enrichment analysis per primer was carried out using the R package TissueEnrich (version 1.14.0)[36] with the default parameters and the $p$ values were adjusted for multiple comparisons with the Benjamini-Hochberg method[37]. The results were visualized with a heatmap generated with the R package ComplexHeatmap (version 2.15.4)[38]. Expressed protein-coding genes per replicate were classified as expressed and not expressed in brain by using as reference the Human Protein Atlas (HPA) list of protein-coding genes detected in brain[25]. For detailed information, R Markdown files are available on github: jumicheel/random_priming (github.com).

### Artificial RNA reverse transcription assay
Low range ssRNA ladder (New England BioLabs, USA) was diluted to 20 ng μl$^{-1}$ and 5 ng μl$^{-1}$, incubated at 90 °C for 3 min, and immediately placed on ice afterwards. 1 μl of the respective dilution was used as input material or SMART-seq3-adapted lysis and reverse transcription as described above in three technical replicates per condition. cDNAs were generated using 0.5 μM of the random 6mer or the random 18mer (Table 1) or a mixture of 0.25 μM of both primers each. cDNAs were pre-amplified as described above and cleaned up using SPRIselect beads in a 3x ratio. After two washing steps with 80% EtOH p.a., the cDNA was eluted in 10 μl nuclease-free water. Samples were quantified by Qubit dsDNA HS and diluted in the same ratio per input amount. Library size distributions of the diluted samples were analyzed with Bioanalyzer using the High Sensitivity DNA kit. Raw data was extracted and analyzed using the R package BioanalyzeR (version 0.10.1)[39].

### Statistical analyses
Statistical tests were performed to analyze significance of the differences between 6mer and 18mer samples detected in the sequencing experiments and significance of the differences between 6mer, 18mer, and the mixture of both primers detected in the electrophoresis experiment. First, normal distribution was tested using the Shapiro-Wilk-Test, and variance equality was tested with the Levene's test using base R stats package (version 4.3.3)[40] and the car R package (version 3.1-2)[41], respectively. Based on the results, the statistical significance test was chosen: two-sided $t$ test (R package rstatix (version 0.7.2)[42]) in case of normal distribution and variance equality or Mann−Whitney $U$ test (base R stats package (version 4.3.3)). Detailed information of all results are reported in Supplementary Data 2. In addition, the statistical test results are summarized in Supplementary Data 1 and

Supplementary Table 4. For each data set, 3 technical replicates starting from input RNA were produced and analyzed. In case of the electrophoresis data, *p*-values were adjusted for multiple comparisons using the Benjamini-Hochberg method[35]. Effect sizes were estimated with Hedges' g statistic using the R package effsize (version 0.8.1)[43].

### Gene function annotation
Genes that were uniquely detected with one primer length in at least two out of three technical replicates (Supplementary Data 3–6 and 8–11) were analyzed for functional enrichment using the DAVID Functional Annotation tool DAVID Version 2021, The DAVID Knowledgebase (v2023q3)[44,45]. ENSEMBL gene IDs were used as input and Homo sapiens as background. All genes that were mapped by DAVID were analyzed for functional enrichment (Count threshold: 2, EASE threshold: 0.1)[44,45]. Enriched pathways with a Benjamini-Hochberg adjusted $p <= 0.05$ were considered as significant.

### Text mining
Quantification of the publications mentioning random primers of different lengths was based on text mining via the PMC Open Access Subset[46]. For all primer lengths between 4 and 24 nucleotides the following search terms were used with the respective nucleotide number: '((("reverse transcription") AND "random tetramer") OR "random 4-mer") OR "random 4mer"'. Data was retrieved on 24 February 2022. Search terms and results are shown in Supplementary Table 6.

### Reporting summary
Further information on research design is available in the Nature Portfolio Reporting Summary linked to this article.

## Data availability
The raw sequencing reads generated in this study have been deposited in the NCBI BioProject database (https://www.ncbi.nlm.nih.gov/bioproject) under accession code PRJNA983129 (Human brain RNA-seq data) and PRJNA1041084 (Vero cell RNA-seq data) The processed sequencing data and electrophoresis data are available at jumicheel/random_priming (github.com). The data shown in this study are provided in the Source Data file. The human genome sequence used in this study is available at http://ftp.ebi.ac.uk/pub/databases/gencode/Gencode_human/release_38/GRCh38.primary_assembly.genome.fa.gz. The human genome annotation used in this study is available at http://ftp.ebi.ac.uk/pub/databases/gencode/Gencode_human/release_38/gencode.v38.primary_assembly.annotation.gtf.gz. ENSEMBL version 105 human gene and transcript annotation (annotation hub 'AH98047') is additionally provided on jumicheel/random_priming (github.com). The AGM genome sequence used in this study is available at http://ftp.ensembl.org/pub/release-105/fasta/chlorocebus_sabaeus/dna/Chlorocebus_sabaeus.ChlSab1.1.dna.toplevel.fa.gz. The AGM genome annotation used in this study is available at http://ftp.ensembl.org/pub/release-105/gtf/chlorocebus_sabaeus/Chlorocebus_sabaeus.ChlSab1.1.105.gtf.gz. The SARS-COV2 genome sequence used in this study is available at http://ftp.ensemblgenomes.org/pub/viruses/fasta/sars_cov_2/dna/Sars_cov_2.ASM985889v3.dna.toplevel.fa.gz. The SARS-COV2 genome annotation used in this study is available at http://ftp.ensemblgenomes.org/pub/viruses/gtf/sars_cov_2/Sars_cov_2.ASM985889v3.101.gtf.gz. The data frame 'brain_all_genes' available at jumicheel/random_priming (github.com) and contains all protein-coding genes detected in the brain from the Human Protein Atlas (downloaded on 11 May 2023). Source data are provided with this paper.

## Code availability
Custom code is available at jumicheel/random_priming (github.com) in the markdown files random_primer_comparison_exp1.Rmd, random_primer_comparison_exp2.Rmd, tissue_enrichment_analysis.Rmd, functional_annotation.Rmd, vero_sars_random_priming.Rmd and artificial_RNA_fragment_analysis.Rmd. In addition, the respective input data files are provided as described in the readme-file.

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

## Acknowledgements

This work was supported by the Ministry for Economics, Sciences and Digital Society of Thuringia under the framework of the Landesprogramm ProDigital (DigLeben-5575/10-9 (J.M.; A.S.; F.A.; D.W.)) and thurAI (2021 FGI 0009 (J.M.; A.S.; F.A.; D.W.)). We thank the Core Facility next-generation sequencing of the Leibniz Institute on Aging—Fritz Lipmann Institute in Jena for their help with Illumina sequencing. We thank Ruben Rose (Institute of Infection Medicine, Kiel University and University Hospital Schleswig-Holstein, Kiel, Germany) and Andi Krumbholz (Institute of Infection Medicine, Kiel University and University Hospital Schleswig-Holstein, Kiel, Germany; Labor Dr Krause und Kollegen MVZ GmbH, Kiel, Germany) for providing RNA from infected Vero cells.

## Author contributions

J.M. and D.W. conceptualized the experiments; J.M. and F.A. performed the experiments; J.M., A.S., and D.W. analyzed the data; J.M., A.S., and D.W. wrote the manuscript; all authors reviewed the manuscript.

## Funding

## Competing interests

The authors declare no competing interests.
