## [Peer Review File · Nature Communications]

Exploring the Impact of Primer Length on Efficient Gene Detection via High-Throughput SequencingReviewers' Comments:

Reviewer #1:

Remarks to the Author:

1. The study aims to reveal the influence of primer length on gene detection using RNA sequencing, especially how it benefits the sequencing from limited input of RNA. But there lacks the evidence if reverse transcription and qPCR efficiency were increased by 18mer primer compared to other primer lengths.
2. It would be more logical if the authors clarify the rationale and significance of optimizing RT primer length in library preparation, and the reason of the selection of 12, 18, and 24mer primer length in the study.
3. With the human brain tissue sample as the objects, the study only showed the characterization of detected genes and proportions mapped reads to various types of RNA. The result would be more substantial if the study supplements the detected longer biotypes, and the GO functional enrichment analysis, etc.
4. How about the performance and efficacy of 18mer primer strategy in detecting short RNA (<200 nucleotides) compared to conventional 6mer primer?
5. What is the effect of reverse transcription with mixed primers of different lengths, and whether is it necessary to verify it?
6. What about the reverse transcription result of these different length primers for different concentration of RNAs (different tissue sources etc), the detailed presentation of the results is preferred.
7. The resolution of part figures is not high enough, and the marks on the axis are not legible
8. More references in the late 3 years would reflect this is up to.

Reviewer #2:

None

Reviewer #3:

Remarks to the Author:

Micheel et al investigate the impact of reverse transcription (RT) primer length on gene detection using RNA-seq. As RT is a foundational step for PCR, qPCR, ddPCR and most RNA-seq analyses, and because it is known to have inefficiencies and create artifacts, implementable improvements to RT have the potential to be very important across a number of fields. Here, the authors clearly show that using an 18-mer RT primer outperforms the standard 6-mer random oligo's normally used. The paper is generally well written and informative. However, there are a number of steps the authors should take to deepen their analysis including by adding statistical rigor to their results before it could be published.

Major issues

1. Is it representative?

Previous literature cited by the authors have shown RT efficiency is highly variable and dependent on many factors. These previous analyses have usually been at the level of 1 or a few genes and have

shown that the amount of RNA you start with and the amount of RT primer used, (among other things), can have large effects on RT efficiency. In the present study the authors tested a single starting amount of total RNA (1ng) and a single concentration of RT primer (0.5uM). The clear question, given the prior literature, is, will the results be representative and reproducible with other input amounts, RT reactions? The amount of starting material (1ng total RNA) is quite low. Many, if not most, RT-PCR & RNA-seq experiments will start with order of magnitude or more sample. Would this result hold when efficiency is not so crucial to gene detection because each gene is represented by a lot more starting RNA? At the very least the authors should cover this thoroughly in the discussion.

A useful feature in method development studies is the use of spike-in controls which provide a known ground truth for gene and isoform detection and quantification. Such controls would have been of use to this study. The authors should, at the least, acknowledge the lack of a ground truth control as one of the limitations.

2. Replicates

It would be helpful if the authors could more clearly explain what replicates were used and how they were made. The manuscript mentions 3 technical replicates for each primer length, but I don't think it is clearly stated what they are replicates of; ie: triplicates of the library preparation starting from total RNA?; triplicate sequencing of 1 library prep etc etc?. I recommend the authors make this clear in the methods and also in the results.

Given this is a methods paper, a methodological overview figure would also be helpful.

For Figure 1E a gene had to be present in 2/3 replicates to be counted. Please specify equivalent information for the other analyses and figures in the manuscript.

For many figures panels it is not clear what "replicates" are being used to generate the data distributions. For example in Fig 1B and C, it is not clear if the distributions represent repeated sub-sampling of different sets of 5 million reads, 3 technical replicates, or both. This also applies to the error bars in fig 1D and similar panels in figure 2, sup figures 2-4. Please specify.

Note that if the distribution in each figure comes from 3 data points/replicates then the value from each replicate should be shown with dots on the figure so the reader can see the underlying data.

Lastly, standard deviations would be more appropriate than SEMs for Fig 1D and Sup Fig 3C.

3. Include statistical tests to backup claims.

Many results would benefit from statistical analyses which are almost entirely absent from the manuscript. For example is the detection of genes across biotype, length, TPM and GC% significantly different between different length primers?

Eg: The statement on line 98-99: "Interestingly, most short biotypes such as snRNAs and snoRNAs were detected most efficiently by the commonly used 6mer primer (Figure 2a)." The difference in the number of features found is quite small and there is a lot of variability, so is there really a difference here?

Eg: The abstract states: "the random 6mer primer showed enhanced detection of shorter RNAs". But from figure 2B it's not clear if it's actually significantly better than longer primers.

4. What are the additional genes found by the 18mer primer?

In a short results section a major piece investigates whether the additional genes detected using the 18mer primer could be artifacts. However it isn't clear (at least to me) how they could be artifacts? I suggest the author explain their reasoning more here. What sort of an artifact could these gene detections be?

Secondly, I'm not convinced that the analysis performed really addresses this question because the authors check their results against the known tissue expression of all detected genes. If one wanted to check the additional 18mer-primer genes were bonafide brain genes, would it not be better to just check these genes specifically?

These comments are related to an overall piece of analysis that also should be better investigated - what are the genes detected only by the 18mer primer?

Sup Fig2 gives a hint that maybe they represent better detection of lowly expressed genes but this result, which could be very important and also relevant to the circumstance under which an 18mer primer could be beneficial, is not sufficiently covered in the manuscript.

5. Data availability

Github link provided is not active

Link to RNA-seq datasets on SRA is not provided.

Minor:

Figure 1E - Does this use sub-sampled data or all data. Please specify.

Line 72 - The authors state there are no "overall differences" in the cDNA library size distributions as shown by the TapeStation. However I don't think any actual analysis has been performed to show/confirm this. In the absence of this I suggest the authors report no "visual" differences.

Line 87 - "While the majority of 5682 genes was detected across all primers". Could be clearer by specifying the denominator. How many genes is this out of?
ie: While the majority (5682 of XXXX) of genes were detected across all primers"....

Line 92 - "Overall, we found that all libraries contain genes of the same biotypes". Could be written better. I think the authors mean all gene biotypes were found in each library.

Lines 156-158. Of previous cited literature the authors state: "Notably, none of these studies examined the effects of RT primer length on a global scale using high throughput sequencing. Instead, they exclusively focused on the detection of specific individual genes."
While the 1st statement is correct, the second is not. Ref 14 Stangegaard et al performed microarrays, which are whole transcriptome experiments, not single gene experiments. Please revise statement.

Line 160-161: "Thus, it is plausible that the observed effects may not be universally applicable to all genes, as we have also observed, but rather dependent on transcript characteristics like length or GC content." This statement is confusing as the authors have themselves claimed that the benefit of 18mer oligo's is depending on transcript length and GC content. Please revise.

Is the cost of using 18mer oligos similar to 6-mers? It would be useful for the authors to comment on this.

Please state the length of the sequencing reads generated in the methods.

Supplementary figure 2B suggests the proportion of expression from each gene biotype was both different and considerably more variable using an 18-mer primer. This doesn't appear to be the case for the deeper replicate experiment but the authors should acknowledge this in the discussion and make some suggestions for what might have caused this.

Second independent experiment - Please specify the max sub-sampled read depth for the replicate experiment (ie: 30 million reads) in the results section so it's clear to the reader.

We would like to thank both reviewers for the thorough and constructive feedback on our manuscript. We strongly believe that the comments and suggestions greatly improved the quality of our manuscript. Here, we summarize the most important additions and changes which overall resulted in 4 additional panels in the existing figures, 3 additional supplementary figures containing a total of 9 new panels as well as 2 new supplementary tables and 9 new supplementary data files compared to the originally submitted manuscript:

- we have added a new RNA-seq dataset derived from Vero cell thus confirming our results using input RNA from a clearly distinct source

- we further confirmed the RT efficiency differences using an RNA ladder as a defined input and quantified the differences on the highly sensitive Agilent bioanalyzer and thereby confirming our results using yet another assay

- we tested the RT efficiency effect across different input concentrations using the RNA ladder assay and confirmed that the results are robust even at larger starting amounts.

- we provide additional analyses on the uniquely detected genes per random primer by pathway enrichment analysis

- we have made numerous clarifying text and figure adaptations, added a methodological overview figure and believe that these changes clearly improved the comprehensibility of the manuscript

Reviewer #1 (Remarks to the Author):

1. The study aims to reveal the influence of primer length on gene detection using RNA sequencing, especially how it benefits the sequencing from limited input of RNA. But there lacks the evidence if reverse transcription and qPCR efficiency were increased by 18mer primer compared to other primer lengths.

This suggestion is certainly interesting. However, we want to emphasize that testing the influence of random primer choice on qPCR would not help to address the primary question of our manuscript: namely to understand the impact of RT primer length on the efficiency of gene detection by RNA sequencing. Gene detection efficiency can unfortunately not be tested by qPCR in particular on a global level. Hence, any outcome of the suggested qPCR experiment would neither support nor undermine any conclusions of our study which is why we believe that this suggestion is beyond the scope of what we tried to find out.

2. It would be more logical if the authors clarify the rationale and significance of optimizing RT primer length in library preparation, and the reason of the selection of 12, 18, and 24mer primer length in the study.

We apologize that the rationale for the primer length choice was unclear. We have now added a paragraph which clarifies the rationale of choice (see new page 8, line 249-254).

*In general, since the effect of primer length choice on gene detection is completely untapped territory any choice larger than 6mer would have been interesting to study. Since the number of possible primers one could have designed is obviously vast, we decided to design primers of lengths that are multiples of 6mer, as it would allow us to cover a relatively large length spectrum in a reasonable scope of a project. We further decided to not test primers of any lengths larger than 24 nt as that would increase the probability to form secondary structures which would hinder transcript priming or enhance the formation of primer dimers (see Hendling & Barišić, *Comput. Struct. Biotechnol. J.* (2019), <https://doi.org/10.1016/j.csbj.2019.07.008>).*

3. With the human brain tissue sample as the objects, the study only showed the characterization of detected genes and proportions mapped reads to various types of RNA. The result would be more substantial if the study supplements the detected longer biotypes, and the GO functional enrichment analysis, etc.

We agree with Reviewer #1 on this point. We have subsequently added multiple new tables (see new Supplementary Data 1 - 4 and 6 - 9) listing all detected genes for any conditions we tested and their respective biotypes in order for the reader to gain a more comprehensive picture of our findings.

As requested, we also performed GO category analyses in order to test for any enrichment of genes that co-cluster into the same functional category (see new Methods section "Gene function annotation" (new page 11), new Supplementary Table 3 and Supplementary Data 5). As expected, no overt functional enrichment that was reproducible across experiments was detected for any of the primers. Importantly, this indicates that primer choice length does not lead to a detection bias for genes of any given category. We believe that this control analysis is an important addition to our manuscript and hence want to thank the reviewer for this suggestion.

4. How about the performance and efficacy of 18mer primer strategy in detecting short RNA (<200 nucleotides) compared to conventional 6mer primer?

The difference between the 6mer and the 18mer regarding RNA length detection is indeed an interesting observation and we agree that it is worth discussing in more detail. We have now expanded on the discussion where we bring up potential explanations for this effect (see new page 6 and 7, lines 207 - 214). Yet, it is important for us to stress that at this point they are rather speculative, since investigating the molecular mechanism of this effect would need more extensive studies in the future.

5. What is the effect of reverse transcription with mixed primers of different lengths, and whether is it necessary to verify it?

Mixing the traditionally used 6mer with the 18mer in order to test if it is possible to combine the advantages of both primers is an interesting idea. In order to test this hypothesis, we have analyzed cDNA generated from a commercially available RNA ladder and tested the reverse transcription efficiency of different length RNA fragments on the bioanalyzer. We used 3 conditions: 6mer only, 18mer only and a 1:1 mix of 6mer and 18mer (see new Supplementary Figure 8 and new Materials and Methods part "Artificial RNA reverse transcription assay" (new

page 11). We again found that the 6mer efficiently transcribes short fragments, while the 18mer demonstrated improved reverse transcription of longer fragments as expected based on our RNA-seq results. Interestingly, the mix of both primers indeed showed efficient reverse transcription of both, long and short fragments. Thus, we believe that mixing 6mer and 18mer primers is likely the optimal setup for future RNA-seq experiments. We want to thank the reviewer for this interesting suggestion which clearly improved our manuscript.

6. What about the reverse transcription result of these different length primers for different concentration of RNAs (different tissue sources etc), the detailed presentation of the results is preferred.

We agree with the reviewer that more conditions would strengthen the generality of our findings. Thus, we have performed the experiments using the RNA ladder as outlined above (Reviewer #1, point 5) using 5 ng and 20 ng as input, which represents 5x and 20x the input amount of the RNA-seq experiment, respectively. Hence, we conclude that our findings are robust at different concentrations.

In addition, we have added RNA-seq results from Vero cells, an African Green Monkey cell line, which represents a tissue source that differs significantly from the originally used human brain lysate. When analyzing the primer effects on the RNA of this source, we found very similar results compared to human brain RNA (see new Supplementary Figures 6 and 7) which demonstrates that the effect of the primer is valid beyond human brain RNA as input. We originally planned to use this dataset as part of a different project studying SARS-CoV-2 infection, however, we understand that demonstrating the effect of our primers using different tissue sources would strongly improve the manuscript and hence decided to add it here.

Together, we believe that these results strongly improve the manuscript as it generalizes our findings beyond the RNA concentrations and tissue sources used in the original manuscript.

7. The resolution of part figures is not high enough, and the marks on the axis are not legible

We are sorry about this technical caveat, as we understand it might have caused inconvenience for the reviewer. Due to Nature Communication's policy we had to embed the figures into the word document which usually improves the readability for the reviewers. But it seems that the figure resolution was reduced. We have now inspected our figures for parts that are hard to read and adapted Figure 1 and Supplementary Figure 4 to enhance the readability of the axis labels.

8. More references in the late 3 years would reflect this is up to.

We have added several publications which intensively investigated the conditions under which RT conditions RNA-cDNA conversion is optimal (Bagnoli et al. 2018, Jia et al. 2021, Hagemann-Jensen et al. 2022, Verwilt et al. 2023, Zucha et al. 2021). These optimization efforts were largely driven by laboratories at the cutting-edge of transcriptomics. We believe that these additions to our manuscript indeed reflect that our efforts are of interest to the transcriptomics community and thus want to thank the reviewer for this suggestion.

Reviewer #3 (Remarks to the Author):

Micheel et al investigate the impact of reverse transcription (RT) primer length on gene detection using RNA-seq. As RT is a foundational step for PCR, qPCR, ddPCR and most RNA-seq analyses, and because it is known to have inefficiencies and create artifacts, implementable improvements to RT have the potential to be very important across a number of fields. Here, the authors clearly show that using an 18-mer RT primer outperforms the standard 6-mer random oligo's normally used. The paper is generally well written and informative. However, there are a number of steps the authors should take to deepen their analysis including by adding statistical rigor to their results before it could be published.

Major issues

1. Is it representative?

Previous literature cited by the authors have shown RT efficiency is highly variable and dependent on many factors. These previous analyses have usually been at the level of 1 or a few genes and have shown that the amount of RNA you start with and the amount of RT primer used, (among other things), can have large effects on RT efficiency. In the present study the authors tested a single starting amount of total RNA (1ng) and a single concentration of RT primer (0.5uM). The clear question, given the prior literature, is, will the results be representative and reproducible with other input amounts, RT reactions? The amount of starting material (1ng total RNA) is quite low. Many, if not most, RT-PCR & RNA-seq experiments will start with order of magnitude or more sample. Would this result hold when efficiency is not so crucial to gene detection because each gene is represented by a lot more starting RNA? At the very least the authors should cover this thoroughly in the discussion.

We highly appreciate the offer to address these concerns through additional discussion especially given that addressing them all through high throughput sequencing would not be financially manageable for us.

However, we still believe that Reviewer #3 raises an important point here. Hence, we still wanted to experimentally test if our main results are stable for higher starting amounts of RNA. We have therefore examined the RT efficiency depending on the RNA amounts by converting a commercially available RNA ladder to cDNA and quantifying the efficiency on the highly sensitive Agilent bioanalyzer. By doing so, we were able to again confirm that 6mer efficiently transcribes short fragments, while the 18mer demonstrated improved reverse transcription of longer fragments as expected from our RNA-seq results and that this result was stable even if one wishes to increase the starting amount by 5x or even 20x (see new Supplementary Figure 8).

Although the general trend in the field of transcriptomics goes strongly towards lower inputs (single cell RNA-seq by and large became the new standard) we believe that these results clearly strengthen the central statements of the manuscript. Moreover, we want to thank the reviewer for inspiring us to come up with yet another assay which again confirmed the central hypotheses of our study.

A useful feature in method development studies is the use of spike-in controls which provide a known ground truth for gene and isoform detection and quantification. Such controls would have been of use to this study. The authors should, at the least, acknowledge the lack of a ground truth control as one of the limitations.

Using spike-in controls is a great idea because of the above mentioned reasons. Unfortunately, commonly used spike-in controls such as the ERCC spike-ins contain only RNA molecules that are 250 nt or longer. Thus we would not be able to adequately compare RT efficiencies of 6mer and 18mer since 6mers show increased efficiency for RNAs that are 200 nt or shorter (see Figure 2b). Therefore, we decided to use a RNA ladder which provides a wider range of RNA lengths at defined concentrations in order to best address this point (see new Supplementary Figure 8 and the new Method section “Artificial RNA reverse transcription assay” (new page 11)).

2. Replicates

It would be helpful if the authors could more clearly explain what replicates were used and how they were made. The manuscript mentions 3 technical replicates for each primer length, but I don't think it is clearly stated what they are replicates of; ie: triplicates of the library preparation starting from total RNA?; triplicate sequencing of 1 library prep etc etc?. I recommend the authors make this clear in the methods and also in the results.

We apologize for not being clear enough regarding the use of replicates on multiple occasions. We have now added the missing information about technical triplicates (which indeed were triplicates of library preparation starting from total RNA) in the methods sections (see new page 8) as well as all figure legends and the results section (see new page 2).

Given this is a methods paper, a methodological overview figure would also be helpful.

We have now created an overview figure that highlights the main variable (RT primer length) which we aimed to investigate in the course of this study (see new Figure 1b). We hope that new figure will help the reader to intuitively understand the central point of our study.

For Figure 1E a gene had to be present in 2/3 replicates to be counted. Please specify equivalent information for the other analyses and figures in the manuscript.

This point is likely related to the misunderstanding caused by the fact that we did not show the data points on top of the box plots, which we have now done. This way it becomes clear that we always used 3 replicates as visible by the number of dots in the plot.

The analyses performed for former Figure 1E (now Figure 1f) as well as Figures 2d, Supplementary Figures 4b and 6b represent exceptions because there we defined primer length-specific genes detected across the technical replicates. We chose to include all genes detected in 2/3 of the replicates since we wanted to also take genes into consideration that

are expressed at lower levels which would largely be left out if we would consider a stricter cutoff of being present in 3/3 samples (see Suppl. Fig 2a).

For many figures panels it is not clear what "replicates" are being used to generate the data distributions. For example in Fig 1B and C, it is not clear if the distributions represent repeated sub-sampling of different sets of 5 million reads, 3 technical replicates, or both. This also applies to the error bars in fig 1D and similar panels in figure 2, sup figures 2-4. Please specify.

We indeed should have been more thorough in our description of the methods. We have now specified the lacking information in the materials and methods (see new page 8) as well as all figure legends and wish to apologize for the inconvenience it would have caused when reviewing our manuscript.

Note that if the distribution in each figure comes from 3 data points/replicates then the value from each replicate should be shown with dots on the figure so the reader can see the underlying data.

As described above, we have now added the data points on top of the boxplots in order to clarify and avoid any misleading impressions for the reader.

Lastly, standard deviations would be more appropriate than SEMs for Fig 1D and Sup Fig 3C.

To the best of our knowledge, the standard error of mean (s.e.m.) is the preferred method of choice for quantifying the error of the mean for technical replicates. Standard deviation (s.d.) in contrast would be appropriate for quantifying the spread of a population which seems like not the right choice in this case, since technical replicates do not inform about population distribution. Thus, we have decided to leave the error quantification as it was before.

However, if there is a gross misunderstanding on our side here, we are perfectly fine with switching to s.d. for error quantification! We just want to be sure that the correct method will be chosen in the end and hope that the reviewer understands our viewpoint here.

3. Include statistical tests to backup claims.

Many results would benefit from statistical analyses which are almost entirely absent from the manuscript. For example is the detection of genes across biotype, length, TPM and GC% significantly different between different length primers?

Eg: The statement on line 98-99: "Interestingly, most short biotypes such as snRNAs and snoRNAs were detected most efficiently by the commonly used 6mer primer (Figure 2a)." The difference in the number of features found is quite small and there is a lot of variability, so is there really a difference here?

Eg: The abstract states: "the random 6mer primer showed enhanced detection of shorter RNAs". But from figure 2B it's not clear if it's actually significantly better than longer primers.

The reviewer raises an important point of discussion here. For each condition that we tested, we used 3 technical replicates since RNA-seq experiments are extremely cost intensive. Our logic was, that rather than increasing the number of replicates in one experiment, we would rather spend the money on a new, completely independent sequencing experiment as this would increase our confidence in the results. The problem with doing statistical tests with “3 against 3” is that 3 samples are considered not enough to reliably inform about the underlying distribution of the data, which is why classically used statistical tests are considered problematic in this case.

Nevertheless, we have performed significance tests as requested that are summarized in the table which can be found attached to our reviewer responses below (page 11 und 12). Additionally, we have tested the assumptions of a T-test such as normality of distribution and homogeneity of variance by performing the Shapiro-Wilk test and Levene's test respectively as we tried to be as careful as possible in this situation. The results of the test demonstrate that almost all comparisons are statistically significant supporting the central hypotheses of the manuscript.

Although the results are in favor of what we like to demonstrate, we hesitate to use them in the manuscript because of the above described reasons. If the reviewer agrees, we would much rather emphasize the fact that the main findings of the manuscript are consistently found across 3 independent RNA-seq experiments (2x human brain RNA-seq, 1x Vero cells RNA-seq which we added as part of this revision) and 1 molecular biology experiment (RT efficiency of artificial RNA). If the reviewer insists on reporting the tests, we will of course implement them in the manuscript and carefully describe the caveats in the methods.

4. What are the additional genes found by the 18mer primer?

In a short results section a major piece investigates whether the additional genes detected using the 18mer primer could be artifacts. However it isn't clear (at least to me) how they could be artifacts? I suggest the author explain their reasoning more here. What sort of an artifact could these gene detections be?

Since application of RT primers of lengths other than 6mer is so vastly understudied we were not sure if priming in those cases would lead to (e.g. fragmented) cDNA products that would be challenging to map computationally. Issues with alignment are known to lead to errors in gene quantification (see Robert & Watson, Genome Biol. (2015), <https://doi.org/10.1186/s13059-015-0734-x>).

Of course it would be a reasonable assumption that this should affect all primers equally but since these experiments were never done using RNA-seq we were simply not sure about that. In all honesty, we just wanted to be on the safe side and thought that this sanity check was worth reporting since it increased our confidence in the results. If the reviewer, however, believes that this control analysis only adds confusion we are open to removing it from the manuscript.

Secondly, I'm not convinced that the analysis performed really addresses this question because the authors check their results against the known tissue expression of all detected

genes. If one wanted to check the additional 18mer-primer genes were bonafide brain genes, would it not be better to just check these genes specifically?

The reason why we have tested all detected genes was because testing the genes that are specific for one primer (e.g. just the 18mer) would exclude any genes that are detected by two or more primers (e.g. 18mer and 12mer or any other combination) but not others. Since the majority of genes would be neglected that way, we decided for a more comprehensive analysis. Nevertheless, we have also conducted the analysis using the 18mer specific genes as requested and found very similar results (see new Figure 2d). If the reviewer agrees, we would leave both analyses in the manuscript, however, we are also open to removing the initial analysis.

These comments are related to an overall piece of analysis that also should be better investigated - what are the genes detected only by the 18mer primer?

Sup Fig2 gives a hint that maybe they represent better detection of lowly expressed genes but this result, which could be very important and also relevant to the circumstance under which an 18mer primer could be beneficial, is not sufficiently covered in the manuscript.

In order to test the interesting hypothesis of increased detection of lowly expressed genes with the 18mer, we needed to perform transcript quantification using the FPKM metric instead of the commonly used TPM. The reason behind this, is that TPM is a relative measure of gene abundance (very much alike percent, but calculating a fraction of a million instead of a hundred). Hence, in a complex library that detects many genes (e.g. due to superior primer such as the 18mer) the TPM values of all genes would be lower compared to a library with the 6mer.

When we performed the analysis we found that the 18mer indeed detected more genes that are expressed at a low level (in particular in the range of FPKM 1-20) as the reviewer suspected based on Supplementary Figure 2. We have added this important result to our manuscript (Supplementary Figure 2b & Supplementary Figure 4e) and want to thank the reviewer for their keen observation that improved our analysis.

5. Data availability

Github link provided is not active

We have now made the repository publicly available.

Link to RNA-seq datasets on SRA is not provided.

All RNA-seq datasets will of course be publicly available once the paper is accepted for publication. For review purposes, we now provide links for our data.

Human brain data (both sequencing runs):

<https://dataview.ncbi.nlm.nih.gov/object/PRJNA983129?reviewer=9hcvihep9lr1ntacd754uop8vg>

Vero cell data:

<https://dataview.ncbi.nlm.nih.gov/object/PRJNA1041084?reviewer=d88ahrki1plmqvrbu6s7f6k7oo>

Minor:

Overall, we want to thank the reviewer for taking the time to examine our manuscript in such great detail. We have now implemented all suggestions.

Figure 1E - Does this use sub-sampled data or all data. Please specify.

In this plot (former Figure 1E, now Figure 1f) subsampled data was used which is now specified in the figure legend.

Line 72 - The authors state there are no "overall differences" in the cDNA library size distributions as shown by the TapeStation. However I don't think any actual analysis has been performed to show/confirm this. In the absence of this I suggest the authors report no "visual" differences.

We have adapted the sentence as suggested.

Line 87 - "While the majority of 5682 genes was detected across all primers". Could be clearer by specifying the denominator. How many genes is this out of? ie: "While the majority (5682 of XXXX) of genes were detected across all primers"....

We have added the missing information for the detected genes.

Line 92 - "Overall, we found that all libraries contain genes of the same biotypes". Could be written better. I think the authors mean all gene biotypes were found in each library.

We have now changed the sentence the following way: "Overall, the same gene biotypes were found in each library."

Lines 156-158. Of previous cited literature the authors state: "Notably, none of these studies examined the effects of RT primer length on a global scale using high throughput sequencing. Instead, they exclusively focused on the detection of specific individual genes." While the 1st statement is correct, the second is not. Ref 14 Stangegaard et al performed microarrays, which are whole transcriptome experiments, not single gene experiments. Please revise statement.

The reviewer is of course right here. This indeed was unfortunate wording on our side, as "these studies" was supposed to refer to the qPCR studies. We have now specified this point in the following way:

“However, neither of the latter two qPCR studies investigated the effect of RT primer length on a global scale. Instead, they focused exclusively on the detection of specific individual genes.”

Line 160-161: "Thus, it is plausible that the observed effects may not be universally applicable to all genes, as we have also observed, but rather dependent on transcript characteristics like length or GC content." This statement is confusing as the authors have themselves claimed that the benefit of 18mer oligo's is depending on transcript length and GC content. Please revise.

We thank the reviewer for this important comment as this was indeed an unfortunate way to phrase it on our part. We have now clarified this point as follows:

“As we have shown in the present study, the RT efficiency with the different primers depends on the transcript length. Therefore, to detect and quantify the effect of primer length on RT, an unbiased, highly sensitive analysis on a global scale using high throughput sequencing, as in our study, is required.”

Is the cost of using 18mer oligos similar to 6-mers? It would be useful for the authors to comment on this.

The cost is indeed similar which we have now mentioned in the discussion (new pages 6 and 7, lines 212 - 214).

Please state the length of the sequencing reads generated in the methods.

The read length of every sequencing experiment is now specified in the Materials section under “Sequencing library preparation”.

Supplementary figure 2B suggests the proportion of expression from each gene biotype was both different and considerably more variable using an 18-mer primer. This doesn't appear to be the case for the deeper replicate experiment but the authors should acknowledge this in the discussion and make some suggestions for what might have caused this.

The reviewer is correct that this variability is indeed not present in the deeper sequenced RNA-seq run, which we now discuss on new page 4, lines 154 - 159.

Second independent experiment - Please specify the max sub-sampled read depth for the replicate experiment (ie: 30 million reads) in the results section so it's clear to the reader.

This information is now provided on new page 4 lines 141 - 142.

Statistical analysis experiment 1

Shapiro-Wilk test	Levenes test	Feature	Group 1 (n=3)	Group 2 (n=3)	Method	p-value, significance	Hedges' g effect size
0.408	0.541	total	6mer	18mer	T-test	0.00973, **	-3.031 (large)
0.241	0.672	protein coding	6mer	18mer	T-test	0.00796, **	-3.21 (large)
0.707	0.556	lncRNA	6mer	18mer	T-test	0.0456, *	-1.873 (large)
0.757	0.768	snRNA	6mer	18mer	T-test	0.0257, *	2.263 (large)
0.902	0.446	snoRNA	6mer	18mer	T-test	0.8, n.s.	0.177 (negligible)
0.181	0.164	pseudogene	6mer	18mer	T-test	0.0367, *	-2.015 (large)
0.242	0.976	0 - 200 bp	6mer	18mer	T-test	0.947, n.s.	0.046 (negligible)
0.386	0.316	200 - 1000 bp	6mer	18mer	T-test	0.0233, *	-2.334 (large)
0.41	0.281	1000 - 2500 bp	6mer	18mer	T-test	0.0264, *	-2.245 (large)
0.323	0.652	2500 - 10000 bp	6mer	18mer	T-test	0.0116, *	-2.883 (large)

Statistical analysis experiment 2

Shapiro-Wilk test	Levenes test	Feature	Group 1 (n=3)	Group 2 (n=3)	Method	p-value, significance	Hedges' g effect size
0.217	0.77	total	6mer	18mer	T-test	0.00146, **	-5.093 (large)
0.214	0.615	protein coding	6mer	18mer	T-test	0.00294, **	-4.225 (large)
0.176	0.657	lncRNA	6mer	18mer	T-test	0.000914, ***	-5.757 (large)
0.756	0.59	snRNA	6mer	18mer	T-test	0.358, n.s.	0.678 (medium)
0.238	1	snoRNA	6mer	18mer	T-test	0.00144, **	5.108 (large)
0.231	0.838	pseudogene	6mer	18mer	T-test	0.00152, **	-5.038 (large)
0.429	0.52	0 - 200 bp	6mer	18mer	T-test	0.00739, **	3.278 (large)
0.99	0.375	200 - 1000 bp	6mer	18mer	T-test	0.647, n.s.	-0.323 (small)
0.557	0.596	1000 - 2500 bp	6mer	18mer	T-test	0.00583, **	-3.504 (large)
0.273	0.231	2500 - 10000 bp	6mer	18mer	T-test	0.0185, *	-2.506 (large)

Reviewers' Comments:

Reviewer #1:

Remarks to the Author:

The revised manuscript is largely improved, and most of the questions are addressed. This study will benefit to RNA related sequencing library construction and it is recommended to publish after minor revision. However, we think the reason for 18mer primer is more efficient than the others need more analysis and discussion; as well as more explain about why the 18mer primer does not have secondary structures produced, et al.

Reviewer #3:

Remarks to the Author:

The authors have done a good job of revising their manuscript and addressed most of my concerns and comments. I think it is important for the authors to now include the results of the statistical tests they have performed and make any necessary revisions based on these. I also have a concern about experiment 2 (human brain at a depth of 30 million reads), which the authors should address.

Unless there is a limit against it the authors may wish to make move some their extensive supplementary figures into the main text. Figure S2 would be a good example.

Important concerns/changes

1. The authors should modify the abstract line that says "the random 6mer primer showed enhanced detection of shorter RNAs." Looking at the figures and the statistical testing results supplied, it is clear this is not a consistent result. I.e: Experiment 1 showed a difference for snRNAs, but not snoRNAs, while experiment 2 had a difference for snoRNAs but not snRNAs.

2. Include statistical tests to backup claims.

In their reply the authors raised concerns about including statistics based on 3 vs 3 replicate RNA-seq results. 3 vs 3 is a standard experimental design in the RNA-seq field due to the cost of sequencing. While more replicates would improve statistical power and measurement accuracy it is standard practice in the field to perform statistical tests on 3 vs 3 data sets. The authors have already performed statistical testing on many of their results (included in their rebuttal only), which was very informative for this reviewer and which should now be included in the manuscript.

To put it simply, the stats results are essential to back-up the various claims made in the manuscript about the differences between data-sets and the benefits of using an 18-mer primer. Without them the manuscript lacks necessary rigor.

Figures and comparison that would benefit from inclusion of statistics are: Fig1C, 1D, 1E (claims of more gene detected at 2.5, 5 and 10 million reads) 2A and 2B, S2B, S2D, S4A, S4B, S4C, S6A, S7A, S7C, S8C, S8D etc.

The authors should also ensure they revise any statements made in the manuscript to be consistent with the results of the statistical testing.

3. Experiment 2

In experiment 2 there is a huge outlier for 1 replicate of the 12mer. It detects far more genes and isoforms than any other sample (S4A,B), mostly likely due to detection of a very large number of lowly expressed genes (S4E). This outlier appears large enough to affect the overall results of this

experiment and so deserves additional scrutiny.

For example - I'd be concerned there is an issue with the read mapping in this sample generating artifacts. I'd recommend the authors take a look at the length and quality of the reads, the results and quality of the mapping, and whether there was something unusual about the cDNA library etc, as something doesn't seem right here. Based on what they find the authors should look to revise Supplementary Figures 4 and 5 and their descriptions of them in the text (lines 139-159).

4. Vero results

Vero cell results show a much weaker benefit of using an 18-mer primer, with a much smaller increase in the number of gene detected. Why? Might an 18-mer primer matter more in a highly transcriptionally complex tissue such as brain?

In addition, the 6, 12 and 24-mer conditions all seem to have 1 outlier replicate which detects a much lower number of genes (see Sup Figure 7A). Is there a technical problem with one set of libraries, or a mapping issue here? Without these outliers it is possible there would be no real difference between the 18-mer primer and the other primer lengths.

The authors should investigate these outliers to determine if there is an issue that can/should be fixed here. Then they should perform statistical analysis of different primer lengths as per human brain experiment 1 and 2 and revise any conclusions based on this. If these results show little, or no, difference between 6mer and 18mer primers the authors should update their manuscript to include a discussion of why this is and the possibility that an 18mer primer is beneficial in some circumstances only.

5. RNA ladder may suggest impact of primer length matters less with higher RNA inputs. Please cover in discussion.

6. Oligonucleotide primer lengths

If I understand table 1 correctly the oligonucleotides being tested are not just a random 6-mer, 12-mer etc. But an oligo containing ~25 specific nucleotides followed by the random priming sequence. I.e.: These are oligo's with "tails" for subsequent steps. I think this creates an important caveat the authors should mention in the discussion. This is that some reverse transcription reactions are performed with primers without tails (from memory Illumina RNA-seq library preps traditionally used tail-less primers). The present of a tail, which does not bind the RNA, may change the binding kinetics of the primer as a whole and could give an 18-mer an advantage over a 6-mer, which might be different or absent in a 6-mer without a tail.

Minor

Regarding SEM vs Standard Deviations (SD). I believe SD's are more appropriate for Figures 1E & S4C for the following reason. What a reader is probably most interested in understanding is not how close to the population mean the measurements are, but how variable the replicates were. I can see the averaged result for 18mers in Fig 1E shows more genes are detected but was this consistent between replicates or highly variable? Standard deviation will show this. The other option would be to plot all replicates on the graph as well as the average, but this might get cluttered. Therefore I would recommend the authors plot standard deviations.

I don't think Sup Fig panels 4E, 5C or 5D are referenced in the text. Please reference and describe.

I recommend the authors be more specific with regards to the cost of a 6mer vs 18mer primer. On line 213 the authors now state using an 18-mer is "cost efficient", but it would be better for the author to make a clear statement of a cost comparison. Ie: "20% extra" or "50% extra" or "50% extra primer cost but negligible overall change in overall cost of RNA-seq".

We would like to thank both reviewers again for their constructive feedback and for recognizing the improvements in our manuscript.

REVIEWER COMMENTS

Reviewer #1 (Remarks to the Author):

The revised manuscript is largely improved, and most of the questions are addressed. This study will benefit to RNA related sequencing library construction and it is recommended to publish after minor revision. However, we think the reason for 18mer primer is more efficient than the others need more analysis and discussion; as well as more explain about why the 18mer primer does not have secondary structures produced, et al.

We thank the reviewer for the positive feedback and for recognizing the relevance of our work to the field of RNA-sequencing library preparation. As suggested, we have now added additional discussion about the yet unknown effect (page 5, lines 220 - 221).

Reviewer #3 (Remarks to the Author):

The authors have done a good job of revising their manuscript and addressed most of my concerns and comments. I think it is important for the authors to now include the results of the statistical tests they have performed and make any necessary revisions based on these. I also have a concern about experiment 2 (human brain at a depth of 30 million reads), which the authors should address.

Unless there is a limit against it the authors may wish to make move some their extensive supplementary figures into the main text. Figure S2 would be a good example.

We agree with the reviewer that Figure S2 would benefit from more visibility and have therefore now added Figure S2 to the main text (now Figure 3).

Important concerns/changes

1. The authors should modify the abstract line that says “the random 6mer primer showed enhanced detection of shorter RNAs.” Looking at the figures and the statistical testing results supplied, it is clear this is not a consistent result. I.e: Experiment 1 showed a difference for snRNAs, but not snoRNAs, while experiment 2 had a difference for snoRNAs but not snRNAs.

This is an important point and we have consequently removed this line from the abstract accordingly. Although we believe that it is likely that the 6mer would

efficiently detect short RNA, we agree that more data e.g. from dedicated short RNA sequencing experiments would make a stronger case.

2. Include statistical tests to backup claims.

In their reply the authors raised concerns about including statistics based on 3 vs 3 replicate RNA-seq results. 3 vs 3 is a standard experimental design in the RNA-seq field due to the cost of sequencing. While more replicates would improve statistical power and measurement accuracy it is standard practice in the field to perform statistical tests on 3 vs 3 data sets. The authors have already performed statistical testing on many of their results (included in their rebuttal only), which was very informative for this reviewer and which should now be included in the manuscript.

To put it simply, the stats results are essential to back-up the various claims made in the manuscript about the differences between data-sets and the benefits of using an 18-mer primer. Without them the manuscript lacks necessary rigor.

Figures and comparison that would benefit from inclusion of statistics are: Fig1C, 1D, 1E (claims of more gene detected at 2.5, 5 and 10 million reads) 2A and 2B, S2B, S2D, S4A, S4B, S4C, S6A, S7A, S7C, S8C, S8D etc.

We extended and included all requested statistical tests (new Supplementary Tables 1 and 5 & detailed information in Supplementary Data 1) and described the analyses in the method section (see page 9 - 10, lines 394 - 406).

The authors should also ensure they revise any statements made in the manuscript to be consistent with the results of the statistical testing.

The following sections of the text have been revised with regards to the results of the statistical test:

-page 1, lines 27 - 28 (abstract)

-page 2, line 68 - 69 (introduction)

-page 2, line 89 (results)

-page 3, lines 106 - 109 and 116 - 118 (results)

-page 4, lines 147 - 149, 152 - 155, 165 - 166 and 174 - 177 (results)

-page 5, lines 191 - 193 (results)

-page 6, line 278 (conclusion)

3. Experiment 2

In experiment 2 there is a huge outlier for 1 replicate of the 12mer. It detects far more genes and isoforms than any other sample (S4A,B), mostly likely due to detection of a very large number of lowly expressed genes (S4E). This outlier appears large enough to affect the overall results of this experiment and so deserves additional scrutiny.

For example - I'd be concerned there is an issue with the read mapping in this sample generating artifacts. I'd recommend the authors take a look at the length

and quality of the reads, the results and quality of the mapping, and whether there was something unusual about the cDNA library etc, as something doesn't seem right here. Based on what they find the authors should look to revise Supplementary Figures 4 and 5 and their descriptions of them in the text (lines 139-159).

During our analyses, we also became aware of the outlier (12mer1). We looked for possible explanations for the emergence of this outlier both in the laboratory part of the work and during the analysis of the sequencing data. When we looked at the standard computational QC metrics (see Table 1 below), we could not find any anomalies. We did, however, find a surprisingly high number of reads mapping to intronic and intergenic regions for the 12mer1 sample (see Figure 1 below). However, we believe that this detail alone is not enough to justify the exclusion of this data (although it clearly would make our data look better). Therefore, we decided to show the full dataset and ultimately advocate for full transparency.

Table 1: Quality control of human brain experiment 2. Initial read length and quality statistics were calculated from the data provided by FastQC. The proportion of uniquely mapped reads was provided by STAR. Mapped read length and quality score were calculated using "samtools stats".

Samples	Initial read length (bp)	Initial mean Phread quality score	Proportion of uniquely mapped reads	Mapped read length (mean; bp)	Mean Phred quality score after mapping
6mer1	2 × 151	34.4	52.3%	127	35.4
6mer2	2 × 151	34.8	62.5%	124	35.6
6mer3	2 × 151	34.7	71.0%	128	35.5
12mer1	2 × 151	34.8	64.8%	128	35.6
12mer2	2 × 151	34.9	85.4%	122	35.8
12mer3	2 × 151	34.9	88.6%	133	35.6
18mer1	2 × 151	34.9	51.0%	125	35.6
18mer2	2 × 151	34.8	46.5%	129	35.5
18mer3	2 × 151	34.6	45.6%	130	35.4
24mer1	2 × 151	34.5	55.7%	130	35.4
24mer2	2 × 151	34.6	49.4%	127	35.6
24mer3	2 × 151	34.7	61.2%	128	35.6

Figure 1: Number of reads mapping to intronic and intergenic regions in human brain experiment 2 as calculated by Qualimap.

4. Vero results

Vero cell results show a much weaker benefit of using an 18-mer primer, with a much smaller increase in the number of genes detected. Why? Might an 18-mer primer matter more in a highly transcriptionally complex tissue such as brain?

The reviewer is right in that, although the trend is clearly similar to the human brain RNA, the 18mer effect is indeed weaker for the Vero cell transcriptome. We believe that this is likely due to missing annotation of the understudied Vero cell transcriptome since many genes might be efficiently transcribed but not quantified in the end.

Although we are convinced that the 18mer effect is general (after all, even the highly different RNA ladder showed a similar effect) we understand that we should be mindful of our wording at this point and have hence adapted the text accordingly (see page 4, lines 174 - 177 and page 6 lines 248 - 250).

In addition, the 6, 12 and 24-mer conditions all seem to have 1 outlier replicate which detects a much lower number of genes (see Sup Figure 7A). Is there a technical problem with one set of libraries, or a mapping issue here? Without these outliers it is possible there would be no real difference between the 18-mer primer and the other primer lengths.

The authors should investigate these outliers to determine if there is an issue that can/should be fixed here. Then they should perform statistical analysis of different primer lengths as per human brain experiment 1 and 2 and revise any conclusions based on this. If these results show little, or no, difference between 6mer and 18mer primers the authors should update their manuscript to include a discussion of why this is and the possibility that an 18mer primer is beneficial in some circumstances only.

We have investigated the above mentioned outliers in detail. As suggested we have again reexamined bioinformatic quality controls (see Table 2 below), but could not find any parameter that correlated with the low gene detection. However, we did find that cDNA concentrations as measured by Qubit directly after RT and preamplification correlated with the number of detected genes (see Table 3 below). This result demonstrates that the low number of detected genes for these outliers is not due to computational issues but rather inefficient reverse transcription reaction itself for those samples. Consequently, it also makes a lot of sense that a primer with superior reverse transition efficiency, such as the 18mer, would be less likely to have such outliers. We have added this interesting point to our discussion (see page 6, lines 250 - 257, new Supplementary Table 5) and thank the reviewer for encouraging a deeper dive into this detail which turned out to be quite informative for us and the reader of the manuscript.

Table 2: Quality control of the Vero cell experiment. Initial read length and quality statistics were calculated from the data provided by FastQC. The proportion of uniquely mapped reads was provided by STAR. Mapped read length and quality score were calculated using “samtools stats”.

Samples	Initial read length (bp)	Initial mean Phread quality score	Proportion of uniquely mapped reads	Mapped read length (mean; bp)	Mean Phred quality score after mapping
6mer1	2 × 251	34.5	80.30%	156	35.8
6mer2	2 × 251	34.5	82.90%	163	35.8
6mer3	2 × 251	34.4	80.40%	158	35.8
12mer1	2 × 251	31	72.20%	215	32.3
12mer2	2 × 251	34.3	77.20%	152	35.8
12mer3	2 × 251	34.3	82.90%	162	35.7
18mer1	2 × 251	34.3	75.40%	154	35.7
18mer2	2 × 251	34.4	78.00%	154	35.8
18mer3	2 × 251	34	71.50%	174	35.2
24mer1	2 × 251	34.5	77.70%	152	35.8
24mer2	2 × 251	34.5	77.20%	154	35.9
24mer3	2 × 251	34.5	81.60%	161	35.8

Table 3: Concentrations of the cDNAs generated after reverse transcription and pre-amplification and quantification of total genes detected per technical replicate.

	6mer		12mer		18mer		24mer	
	ng/ μ l	total genes detected	ng/ μ l	total genes detected	ng/ μ l	total genes detected	ng/ μ l	total genes detected
replicate 1	7.92	7695	11.9	7888	13.2	8585	8.92	8015
replicate 2	2.90	6086	6.92	7649	5.86	7586	7.44	7847
replicate 3	7.22	7498	3.44	6491	7.72	7843	2.84	6443

5. RNA ladder may suggest impact of primer length matters less with higher RNA inputs. Please cover in discussion.

The reviewer mentions an interesting point and we agree that it is worth elaborating on this in the discussion. Thus, we have now addressed this point in a new paragraph of the discussion (page 6, lines 242 - 247).

6. Oligonucleotide primer lengths

If I understand table 1 correctly the oligonucleotides being tested are not just a random 6-mer, 12-mer etc. But an oligo containing ~25 specific nucleotides followed by the random priming sequence. I.e.: These are oligo's with "tails" for subsequent steps. I think this creates an important caveat the authors should mention in the discussion. This is that some reverse transcription reactions are performed with primers without tails (from memory Illumina RNA-seq library preps traditionally used tail-less primers). The present of a tail, which does not bind the RNA, may change the binding kinetics of the primer as a whole and could give an 18-mer an advantage over a 6-mer, which might be different or absent in a 6-mer without a tail.

We thank the reviewer for the suggestion and have added a clarifying paragraph to the discussion (page 6, lines 265 - 272).

Minor

Regarding SEM vs Standard Deviations (SD). I believe SD's are more appropriate for Figures 1E & S4C for the following reason. What a reader is probably most interested in understanding is not how close to the population mean the measurements are, but how variable the replicates were. I can see the averaged result for 18mers in Fig 1E shows more genes are detected but was this consistent between replicates or highly variable? Standard deviation will show this. The other option would be to plot all replicates on the graph as well as the

average, but this might get cluttered. Therefore I would recommend the authors plot standard deviations.

As we mentioned before, we are open to this change and believe that the reviewer made a reasonable case. Therefore, we have now changed the variation quantification to Standard Deviations for the requested figures and have updated the figure legends accordingly.

I don't think Sup Fig panels 4E, 5C or 5D are referenced in the text. Please reference and describe.

We apologize that we overlooked this. The respective passages have now been added to the results section:

former Supplementary Figure 4e, now 3e: page 4, lines 152 - 155

former Supplementary Figure 5c, now 4c and former Supplementary Figure 5d, now 4d: page 4, lines 162 - 166.

I recommend the authors be more specific with regards to the cost of a 6mer vs 18mer primer. On line 213 the authors now state using an 18-mer is "cost efficient", but it would be better for the author to make a clear statement of a cost comparison. I.e: "20% extra" or "50% extra" or "50% extra primer cost but negligible overall change in overall cost of RNA-seq".

We have added more detailed information on the primer costs as suggested (see page 5, lines 225 - 229). Although we gladly add this information it is important for us to caution the reader that these numbers are very much dependent on variables such as region or vendor and will likely change in the future.

Reviewers' Comments:

Reviewer #1:

Remarks to the Author:

Thanks to the hard work of the Editors and Authors, I have no further comments on the present manuscript.

Reviewer #3:

Remarks to the Author:

The authors have once again done a good job in addressing concerns. I suggest the following to complete the revisions:

1. The use of a supplementary table to show statistical results is less useful than including the results as part of the figures (which is standard practice). I'd recommend the latter.

Also note the some labeling in Sup Table 1 is incorrect (ie: Human brain expt 2 stats labelled as coming from Sup Figure 1A) and the order of the results jumps around between figures instead of processing logically.

2. Creating main figure 3 is good. However, it also leads to figures being covered out of order. Rearranging panels between figures would fix this.

3. Artificial RNA experiment.

Statistical testing (Sup Table 5) showed no difference between 6-mer and 18-mer primers at 20ng RNA input (compared to 5ng where there was a difference). However the authors state in line 195: "the effect was observed both with 5ng input RNA and with 20 ng input RNA". This claim should be corrected.

The related claim in the discussion (line 245-246) that the effect at 20ng "might not be quite as pronounced" should also be corrected. The authors should simply state the difference was not significant.

4. VERO cells

Similar to point 3 above, statistical testing (Sup Table 1) showed no difference between 6-mer and 18-mer primers in the number of genes detected. On line 176-177 the authors state the differences are "not as big as in the human brain experiments". This implies a difference was observed in the VERO cells. They should instead state the differences were not significant.

Similarly in the discussion line 248 the authors state "the Vero cell experiment showed a lower positive effect of the 18mer primer compared to the human brain experiments". The lack of a significance difference means no positive effect was observed and this claim should likewise be corrected.

We thank again both reviewers for their constructive feedback throughout the review process and for acknowledging our efforts to improve the manuscript.

REVIEWERS' COMMENTS

Reviewer #1 (Remarks to the Author):

Thanks to the hard work of the Editors and Authors, I have no further comments on the present manuscript.

Reviewer #3 (Remarks to the Author):

The authors have once again done a good job in addressing concerns.

I suggest the following to complete the revisions:

1. The use of a supplementary table to show statistical results is less useful than including the results as part of the figures (which is standard practice). I'd recommend the latter. Also note the some labeling in Sup Table 1 is incorrect (ie: Human brain expt 2 stats labelled as coming from Sup Figure 1A) and the order of the results jumps around between figures instead of processing logically.

The results of the statistical tests are included in the figures now. Supplementary Table 1 including the summary of statistical tests and their results has been corrected and updated in accordance with the rearranged panels.

The order of Supplementary Table 1 indeed does not reflect the order in which the data is mentioned in the manuscript. However the order follows an internal logic since it enables direct comparisons across individual experiments and in that way represents a more useful addition to the manuscript in comparison to merely replicating the statistical results that can be obtained from the figures .

2. Creating main figure 3 is good. However, it also leads to figures being covered out of order. Rearranging panels between figures would fix this.

The Figure panels have now been rearranged so that they are displayed in the order in which they are mentioned in the manuscript.

3. Artificial RNA experiment.

Statistical testing (Sup Table 5) showed no difference between 6-mer and 18-mer primers at 20ng RNA input (compared to 5ng where there was a difference). However the authors state in line 195: "the effect was observed both with 5ng input RNA and with 20 ng input RNA". This claim should be corrected.

We thank the reviewer for highlighting these inconsistencies and changed these sentences accordingly:

"We further tested if different RNA input amounts would affect this result and found that the effect was observed with 5 ng input RNA ($p = 0.00737$) (Supplementary Figure 7a, 7c) and to a smaller extent with 20 ng input RNA (not significant) (Supplementary Figure 7b, 7d)."

The related claim in the discussion (line 245-246) that the effect at 20ng "might not be quite as pronounced" should also be corrected. The authors should simply state the difference was not significant.

Changed to:

"Increasing the input level in the fragment size experiment to 20 ng shows a lower, statistically insignificant effect compared to the random 6mer, which may indicate that the priming efficiency of the different primer lengths converges at higher input levels."

4. VERO cells

Similar to point 3 above, statistical testing (Sup Table 1) showed no difference between 6-mer and 18-mer primers in the number of genes detected. On line 176-177 the authors state the differences are "not as big as in the human brain experiments". This implies a difference was observed in the VERO cells. They should instead state the differences were not significant.

We agree with the reviewer that the sentence could be understood as a statistically significant change. Accordingly, we have changed the sentence to make it clear that only a trend is described which does not reach statistical significance. Especially regarding the uniquely detected genes (Supplementary Figure 5b), we see for example that the random 18mer and the other longer primers perform better than the random 6mer and we would like to mention this.

new sentence: "Again, we see that there is a tendency towards enhanced gene detection with longer random primers but the overall differences between the primers are not as big as in the human brain experiments and do not reach statistical significance (Supplementary Table 1)."

Similarly in the discussion line 248 the authors state "the Vero cell experiment showed a lower positive effect of the 18mer primer compared to the human brain experiments". The lack of a significance difference means no positive effect was observed and this claim should likewise be corrected.

Changed to:

"In addition, the Vero cell experiment showed a lower positive effect of the 18mer primer compared to the human brain experiments which did not reach statistical significance (Supplementary Table 1)."